# Anti-Cancer Activity of the Combinational Treatment of Noozone Cold Plasma with p-FAK Antibody-Conjugated Gold Nanoparticles in OSCC Xenograft Mice

**DOI:** 10.3390/biomedicines10092259

**Published:** 2022-09-12

**Authors:** Jeong-Hae Choi, Hee-Jin Gu, Kwang-Ha Park, Dae-Seok Hwang, Gyoo-Cheon Kim

**Affiliations:** 1Corporate Affiliated Research Institute, Feagle Co., Ltd., Yangsan 50614, Korea; 2Department of Oral and Maxillofacial Surgery, School of Dentistry, Pusan National University, Yangsan 50612, Korea; 3Department of Oral Anatomy and Cell Biology, School of Dentistry, Pusan National University, Yangsan 50612, Korea

**Keywords:** oral squamous cell cancer (OSCC), no-ozone cold plasma (NCP), gold nanoparticles conjugated with p-FAK antibody (p-FAK/GNP)

## Abstract

Oral squamous cell cancer (OSCC) is the most common type of oral cancer (about 80–90% of cases) and various research is being done to cure the disease. This paper aims to verify whether treatment with no-ozone cold plasma (NCP), which is designed for safe usage of the plasma on oral cavities, in combination with gold nanoparticles conjugated with p-FAK antibody (p-FAK/GNP) can trigger the selective and instant killing of SCC-25 cells both in vitro and in vivo. When SCC25 and HaCaT cells are exposed to p-FAK/GNP+NCP, the instant cell death was observed only in SCC25 cells. Such p-FAK/GNP+NCP-mediated cell death was observed only when NCP was directly treated on SCC25 harboring p-FAK/GNP. During NCP treatment, the removal of charged particles from NCP using grounded electric mesh radically decreased the p-FAK/GNP+NCP-mediated cell death. This p-FAK/GNP+NCP-mediated selective cell death of OSCC was also observed in mice xenograft models using SCC25 cells. The mere treatment of p-FAK/GNP and NCP on the xenograft tumor slowly decreased the size of the tumor, and only about 50% of the tumor remained at the end of the experiment. On the other hand, 1 week of p-FAK/GNP+NCP treatment was enough to reduce half of the tumor size, and most of tumor tissue had vanished at the end. An analysis of isolated tissues showed that in the case of individual treatment with p-FAK/GNP or NCP, the cancer cell population was reduced due to apoptotic cell death. However, in the case of p-FAK/GNP+NCP, apoptotic cell death was unobserved, and most tissues were composed of collagen. Thus, this paper suggests the possibility of p-FAK/GNP+NCP as a new method for treating OSCC.

## 1. Introduction

Oral cancer is one of the 10 main cancers and is a major health problem [1] due to its high lethality (about 50%) [2]. Oral cancer is often called oral squamous cell carcinoma (OSCC), since 90% of oral neoplasms are OSCC [3]. The cause of the high lethality of OSCC is that its diagnosis is mainly done in the advanced stage [4]. Despite the recent development of medical technologies, the death rate of OSCC has not improved. The most efficient treatment of OSCC is early discovery and its removal through surgery. However, not only is early discovery very difficult, if OSCC occurs in parts where surgery is impossible, the surgical approach is unhelpful [5]. To overcome the limitations of such surgical approaches, chemotherapy using various anti-cancer drugs along with radiotherapy is recommended [6] and, recently, research on methods involving immunotherapy and gene therapy have been done [7,8]. However, despite various scientific efforts, OSCC lethality is still high. Thus, new methods to treat OSCC are desperately needed.

For the last 20 years, various research involving the medical application of physical plasma have been conducted [9]. Plasma is a physical term which refers to the fourth state of matter created by applying energy to a gas and it is composed of positive gaseous ions and electrons. However, in the process of plasma generation, light, heat and reactive species can be generated along with ions and electrons, and these elements can trigger several chemical and physical reactions in its target [10]. Recently, medical devices generating plasma below 42 °C, called non-thermal plasma or cold atmospheric plasma (CAP), have been under development [11]. Various biological benefits of CAP, such as sterilization [12], anti-inflammation [13,14], wound healing [15] and tissue recovery [16,17] have been presented. The anti-cancer activity of CAP is being actively researched too [18]. According to Yan et al., anti-cancer effects of CAP on around 20 cancer types, including brain cancer, lung cancer, colorectal cancer, skin cancer and breast cancer have been reported [19]. Among these cancer types, the research presenting the beneficial effects of CAP on head and neck cancer only account for 3.28%, and research on the anti-cancer effect of CAP on OSCC is rare. Ramireddy et al. reported that helium plasma jet treatment on OSCC causes apoptotic cell death due to the increase of oxidative stress in tumor cells [20]. In the case of Lee et al., it has been reported that the treatment of OSCC with nitrogen-based plasma jets induced apoptotic cell death, and they also proposed the increased intracellular level of reactive oxygen species (ROS) as a reason for this phenomenon [21]. However, most of these studies are the result of the direct treatment of CAP into the OSCC cell line, and the anti-OSCC activity of CAP in in vivo experiments has not yet been confirmed. In addition, the increase in intracellular ROS by CAP treatment and the resulting apoptotic cell death is essential for the anti-cancer efficacy of CAP, and these phenomena are similar to those in cancer cells exposed to various cancer treatment methods such as anti-cancer drugs and radiotherapy [22,23]. As with most cases of drugs and treatment technologies that induce apoptotic cell death through increased ROS inside cancer cells, CAP also needs to enhance its selectivity against cancers. In addition, as shown in the report by Song et al., cell death via CAP is rarely observed in the oxidative stress-resistant cancer cells [24], and a method to amplify the anti-cancer effect of CAP is still needed.

In our previous study, we confirmed that when the G361 human melanoma cells and HaCaT human keratinocytes were treated with CAP and gold nanoparticles conjugated with p-FAK antibodies (p-FAK/GNP) in combination, only G361 cells were killed immediately [25]. However, this research was not only limited to the in vitro level, but also needs additional studies to identify the specific mechanism of action. In addition, since the CAP used in this study was a CAP using air in the atmosphere, there was a limitation that it cannot be used in the oral cavity because a large amount of ozone is inevitably generated. Therefore, in this study, the anti-OSCC effect of the combinational treatment of p-FAK/GNP and no-ozone cold plasma (NCP), which is a plasma designed to be safely used in the mouth, was verified. Also, we tried to identify the specific mechanism of action by performing several in vitro experiments. In addition, we examined the possibility that the combination of NCP and p-FAK/GNP can be applied to OSCC treatment using xenograft mice models.

## 2. Materials and Methods

### 2.1. Cell Culture

All cells used in this study (SCC25, YD-10B, MG63, Hs68 and HaCaT cells) were cultured in Dulbecco’s Modified Eagle’s Medium:Nutrient Mixture F-12 (DMEM/F12) Media (1:1) and DMEM (Hyclone, Salt Lake City, UT, USA) media, respectively. Cultures were maintained at 37 °C and 5% CO_2_ atmosphere in a humidified incubator. When the cell confluence reached about 80–90% in a 100 mm dish, the cells were washed with phosphate buffer saline (PBS) twice, sub-cultured using trypsin-EDTA (Gibco, Grand Island, NY, USA), and the media was changed every 2–3 days.

### 2.2. Immunofluorescence

The cells were seeded in a confocal dish at a density of 250 cells/dish and incubated for 48 h. The cells were fixed for 30 min with 4% paraformaldehyde at room temperature and incubated with p-FAK antibody overnight at 4 °C. Then, the cells were incubated with goat Alexa 488 anti-mouse IgG (Invitrogen, Carlsbad, CA, USA) for 2 h. The nuclei of the cells were stained with 4′,6-diamidino-2-phenylindole (DAPI). Fluorescent images were taken with Zeiss LSM 510 laser scanning confocal microscope (Goettingen, Germany).

### 2.3. Western Blot Analysis

Western blot analysis against p-FAK protein using the protein extracts of oral cancer cells (SCC25, YD-10B and MG63) and normal cells (HaCaT, Hs68) was performed. When the confluence of each cell reached 90% in a 100 mm cell culture dish, the cells were washed with PBS twice and incubated with cell lysis buffer (10mM Tris/HCl, pH7.2, 1% Triton X-100, 150 nM NaCl) for 30 min at 4 °C. The lysate was clarified by centrifugation at 12,000 rpm for 15 min at 4 °C, and the supernatant was obtained. The protein content of the lysate was measured using a Bradford protein assay. About 30 μg of protein lysates from each cell lysate were resolved by size using 8% sodium dodecyl sulfate polyacrylamide gel electrophoresis (SDS-PAGE) and transferred to the PVDF membrane. The membrane was blocked with 5% skim milk for 1 h and incubated with anti p-FAK (Biotechnology, Santa Cruz, CA, USA) overnight at 4 °C. After that, the membrane was incubated with anti-mouse IgG for 1h. The membrane was treated with ECL detection solution, and the protein bands were detected using Alpha Imager HP (Alpha Innotech, San Leandro, CL, USA).

### 2.4. Formation of p-FAK/GNP Conjugates and Its Treatment

The GNP was conjugated with p-FAK antibody, as described in our previous report [25]. Briefly, 11-mercaptoundecanoic acid (MUA) (0.1 mg/mL) was added on gold colloid (30 nm) and then incubated at 4 °C overnight. Then, 1 mM of N-hydroxysuccinimide (NHS) and 1 mM of N-ethyl-N’-(3- dimethylaminopropyl) carbodiimide (EDC) was added onto the MUA−gold nanoparticles and incubated at 4 °C for 20 min. After that, this mixture sample was centrifuged at 4 °C and 13,000 rpm for 15 min. After the removal of the supernatants, a PBS buffer (1 mM, pH7.0) was added to form a GNP solution and mixed with p-FAK antibodies at a 1:1 ratio. After incubating this p-FAK/GNP solution at 4 °C for 2 h, the solution was diluted with the culture media 5 times, treated on the cells and incubated for 2 h.

### 2.5. NCP Device

In this study, the cells and the xenograft mice were treated with the NCP device, which were described in our previous report [26,27]. In brief, this device is composed of a handheld plasma generating part (handpiece), and the main body consists of a switched mode power supply, solenoid valve, gas flow rate controller, main board and high voltage board. The main body is connected to the handpiece by a cable that is approximately 1 m in length, and the high voltage signal and argon gas are transferred from the main body to the handpiece through the cable. Approximately 3 kVpp of high voltage with a frequency of 20 kHz is transferred from the high voltage circuit inside the main body to the plasma generator in the handpiece, while the argon gas flows at a fixed rate of 1 slm (standard liter per minute). The plasma power at this condition was about 0.68 W.

### 2.6. Live/Dead Cell Staining

The viability of cells after the individual treatment of NCP and p-FAK/GNP and its combination was evaluated using live/dead viability/cytotoxicity assay kit (Invitrogen Grand Island, NY, USA). The SCC25 and HaCaT cells seeded in a confocal dish were treated with NCP (for 5 min) and p-FAK/GNP (for 1 h). For the combination treatment, the cells were pre-treated with p-FAK/GNP for 1 h and then the non-specific bound of p-FAK/GNP was removed by washing with PBS, and then treated with NCP in the presence of the fresh media. Immediately after the treatment, the cells were treated with a calcein AM and ethidium homodimer-1 mixture and reacted together for 20 min at 37 °C. The live (green) and dead (red) cells were detected using the Zeiss LSM 700 laser scanning confocal microscope (Goettingen, Germany).

### 2.7. Animal Experiments

Five-week-old BALB/c nude mice were purchased from Orient-Bio Inc. (Seoul, Korea) and adapted in the experimental animal facility of Pusan National University School of Medicine. Mice were homed for a week in a barrier cage with room temperature 23 ± 2 °C, humidity 55 ± 5%, ventilation 10~15 times/h and light and shade in 12 h increments, and water and feed were provided in a self-regulating manner.

This animal experiment was performed in accordance with the regulations of the Animal Experimentation Ethics Committee with the approval of the Animal Experimentation Ethics Committee of Pusan National University (PNU-2019-2342).

### 2.8. Anti-OSCC Activity Evaluation in a Mice Xenograft Model

To evaluate the anti-OSCC activity in vivo, mice xenograft models using SCC25 cells were adopted. The cells in a concentration of 1 × 10^6^ cells/mL in PBS were mixed with Matrigel in a 1:1 ratio, and 150 μL of this mixture was injected into the back of each nude mouse to induce tumors. After a 3-week tumorigenic period (tumor size reached ≥5 mm diameter), the mice were divided into 4 groups (non-treated, NCP, p-FAK/GNP and p-FAK/GNP+NCP, n = 4). Then, the injection of p-FAK/GNP was performed once a week and NCP treatment was performed 3 times a week for 4 weeks. The non-treated group and NCP group were injected with PBS once a week at the same time for 4 weeks. During the experiment period, the general condition of mice was observed, and tumor size (length, width and height) was measured using Vernier caliper once a week. The tumor volumes were calculated as follows: length × width × height × 0.52, according to the report of Ni et al. [28]. After the last treatment, tumor tissues were isolated and measured by weight, and then used for histological analysis.

### 2.9. Histological Assays

The tumor tissue slice with 5 μm thickness was deparaffinized using xylene and rehydrated using ethanol. Then, the tissues were washed with PBS and stained with heamtoxylin and eosin (H&E) and Masson’s trichrome. After the staining, morphological characteristics were analyzed using optical microscopy. Immunohistochemistry using an antibody against cleaved caspase3 (Cell Signaling, Danvers, MA, USA) was performed to visualize apoptotic cell death in the tissues. After deparaffinization and rehydration, the tissues were reacted with anti-cleaved caspase 3 antibodies at 4 °C overnight. After washing, the cleaved caspase 3 protein was visualized through Rabbit specific HRP/DAB detection IHC kit (Abcam), and the nuclear was counterstained using hematoxylin. To take photographs of the IHC, Zeiss Axio scan Z1 Digital Fluorescence slide scanner (Goettingen, Germany) was used. To detect proliferating cells in the tumor tissues, an immunofluorescence assay using anti-Ki67 antibodies (Abcam, Cambridge, UK) was performed. The tissues were reacted with anti-Ki67 antibodies at 4 °C overnight and then the tissues were reacted with anti-rabbit IgG for 1 h. The nuclear was stained with propidium iodide (PI). The results were confirmed using a Zeiss LSM 700 laser scanning confocal microscope (Goettingen, Germany).

### 2.10. Statistical Analysis

Data analysis in this study was conducted using IBM SPSS Statistics 20 (SPSS, Chicago, IL, USA) and expressed as means ± standard errors (SE). The statistical differences were calculated utilizing a one-way ANOVA. The * symbols between bars indicate statistically significant differences between groups (*p* values < 0.05, one-way ANOVA with Duncan’s post hoc test).

## 3. Results

### 3.1. Phosphorylated FAK Protein Was Highly Expressed in OSCC Cells

First, the applicability of antibodies targeting phosphorylated FAK proteins was examined as a method for selectively targeting GNP to OSCC. The expression of p-FAK protein in MG63 cells, an osteosarcoma cell that can occur frequently in the oral cavity, as well as two OSCC cell lines, SCC25 and YD-10B cells, was compared with the expression of HaCaT (human keratinocyte) and Hs68 (Figure 1). As a result, higher p-FAK protein expression was observed in MG63 cells along with SCC25 cells and YD-10B cells compared to normal cell lines.

Immunofluorescence assay against p-FAK protein using HaCaT cells and SCC25 cells was performed to visualize the expression pattern of p-FAK within normal and OSCC cells once again (Figure 1B). As a result, p-FAK protein was observed around the nucleus in both cells, but it was confirmed that the florescence intensity of SCC25 cells was much higher than HaCaT cells.

### 3.2. The Combinational Treatment of NCP with p-FAK/GNP Selectively Kills SCC25 Cells Immediately after Treatment

Antibodies targeting p-FAK proteins overexpressed in SCC25 cells were conjugated to GNP according to the method established in our previous studies to produce p-FAK/GNP (Figure 2A). In this study, the handpiece of the NCP device was fixed using a stand clipper, and the plasma gas outlet was kept at 1 cm from the cell during the cell treatment (Figure 1B). Live and dead assays were performed to see if the combination of p-FAK/GNP and NCP could selectively induce immediate and selective cell death of OSCC cells. For this, HaCaT cells and SCC25 cells were divided into 4 groups; non-treated control, p-FAK/GNP (cultured for 2 h after p-FAK/GNP treatment), NCP (treated for 5 min), p-FAK/GNP+NCP (incubation for 2 h after p-FAK/GNP treatment, washing and NCP). As a result of the live and dead assays performed immediately after NCP treatment (Figure 2C), p-FAK/GNP single treatment and NCP single treatment only caused a change in the shape of the live cells (green) of HaCaT cells and SCC25 cells, but no dead cells (red) were observed. On the other hand, when pFAK/GNP and NCP were treated in combination, dead cells were not observed in HaCaT cells, whereas the decrease of live cells and the increase of dead cells were detected in SCC25 cells.

### 3.3. The Effects of Plasma-Activated Media Treatment on p-FAK/GNP-Treated SCC25 Cells Was Weaker Than the Direct NCP Treatment

To elucidate how NCP treatment stimulated p-FAK/GNP to induce the immediate death of SCC25 cells, we used two different NCP treatment methods and tested the changes in the effect of NCP on SCC25 cells according to the treatment method. As illustrated in Figure 3A, the p-FAK/GNP+NCP treatment method is a direct treatment method in which cells pre-treated with p-FAK/GNP are directly exposed to NCP for 5 min. On the other hand, in the p-FAK/GNP+PAM method, the cell-free medium was treated with NCP for 5 min to create plasma-activated media (PAM), and then PAM was added into the cells pre-treated with p-FAK/GNP and cultured for 5 min. In this method, only changes in the liquid medium induced by NCP can stimulate p-FAK/GNP. Looking at the immediate SCC25 cell death efficacy according to two NCP treatment methods, it was confirmed that the p-FAK/GNP+NCP treatment method induced cancer cell death more effectively than the p-FAK/GNP+PAM method (Figure 3B).

### 3.4. The Placement of Electronic Grounded Mesh during NCP Treatment Hindersp-FAK/GNP+NCP-Mediated SCC25 Cell Death

Additional experiments were performed to narrow down the possible candidate among various working elements of NCP for the induction of p-FAK/GNP+NCP-mediated immediate cell death. For this, three different NCP treatment methods were used as described in Figure 4A. The first method was a direct treatment method where pretreated p-FAK/GNP SCC25 cells were treated with NCP for 5 min (p-FAK/GNP+NCP). The p-FAK/GNP+NCP-DE method was a method treating NCP in the presence of a mesh composed with de-electric material between the NCP device and the cells. Finally, in the p-FAK/GNP+NCP-EG method, a copper-based grounded mesh was placed between cells and NCP devices during the treatment. The results showed (Figure 4B) that in samples treated with the p-FAK/GNP+NCP method, large numbers of cells were intensively killed in one site, while in the case of de-electric (DE) mesh installation, the distribution of dead cells were sporadic with a slight decrease in the number of dead cells. Interestingly, it was confirmed that the immediate killing efficacy by NCP was significantly inhibited by the installation of the electric grounded (EG) mesh. The average number of dead cells observed on one screen after three repeated experiments showed that about 39 cells were killed in a simple NCP-treated sample, while about 28 cells were killed in a p-FAK/GNP+NCP-DE sample, and in the case of p-FAK/GNP+NC-EG, only about 10 dead cells were monitored (Figure 4C).

### 3.5. The Synergistic Anti-Tumor Effect of the Combinational Treatment of NCP and p-FAK/GNP in Mice Xenograft Models

To confirm that the anti-OSCC effect of the combination treatment of NCP and p-FAK/GNP identified in the in vitro model can be reproduced in in vivo models, a mice xenograft model was adopted. As shown in Figure 5A, SCC25 cells were injected into nude mice to form a tumor for 3 weeks, and then divided into non-treated, p-FAK/GNP, NCP and p-FAK/GNP+NCP groups and treated for 4 weeks as described. As Figure 5B shows, the tumor size gradually decreased compared to non-treated controls in mice treated with NCP and p-FAK/GNP alone. On the other hand, in the p-FAK/GNP+NCP group, the size of the tumor rapidly decreased after 1 week of treatment and bruised skin was observed. Thereafter, the tumor size of the p-FAK/GNP+NCP group gradually decreased from week 1 to the end of the treatment. Looking at the change in the external size of the xenograft tumor summarized in Figure 5C, both NCP and p-FAK/GNP showed a gradual reduction in the size of the tumor, with a statistically significant difference of about 55% compared to the pre-treatment size after 3 weeks, and a tumor about 41% of its original size remained after 4 weeks. On the other hand, in the experimental group treated with p-FAK/GNP and NCP in combination, only about 45% of the tumor size remained after 1 week of treatment, indicating statistically significant efficacy, decreasing to 38% after 2 weeks of treatment, 25% after three weeks of treatment and only about 13% of the tumor remained at the end of the treatment.

### 3.6. The Tissue Isolated from the Mice Treated with pFAK/GNP in Combination with NCP Was Not a Tumor

After all treatment procedures were completed, the tumor tissues were isolated from the mice and the changes in the size, shape and weight of the tumor according to each treatment method were examined. As shown in Figure 6A, the tumor in the non-treated group was the largest and denser than in the other 3 groups, and the size of tumor tissues from the NCP or p-FAK/GNP groups was reduced compared to non-treated controls. Interestingly, it was confirmed that the size of tumor tissues from the p-FAK/GNP+NCP group was the smallest among 4 groups, and the tissue was transparent. When comparing the weight of the tumor tissues of 3 experimental groups to control groups, the average weight of the NCP group was 45% of the control, the p-FAK/GNP group was 53% of the control and the weight of the p-FAK/GNP+NCP group was 38% of the control. However, there were no significant differences between the 3 experimental groups.

To examine the histological changes of the tumor by each treatment method, tissue analysis using H&E and Masson’s T staining was first performed (Figure 6B). First, the results of H&E staining showed that in the tumor of non-treated control groups, most of the tissues were composed of cells and observed with keratinization, a characteristic of SCC tumors, while the density of cells decreased somewhat in the tumors of the NCP and p-FAK/GNP groups. Surprisingly, the tissues isolated from p-FAK/GNP had few cells, and the keratinization phenotype was not detected. The results of Masson’s T staining results also showed that most parts of non-treated group tumor tissues were composed of cells, resulting in little collagen. However, in the case of the NCP and p-FAK/GNP groups, decreased cell density and slightly increased tissue collagen content was monitored. Interestingly, it was confirmed that most parts of the tissues isolated from p-FAK/GNP+NCP groups were composed of collagen rather than cells.

### 3.7. Apoptotic Cell Death Was Not Monitored in the Tumors Treated with NCP and p-FAK/GNP in Combination

To examine how the single treatment of NCP and p-FAK/GNP and its combination showed anti-cancer effects, we first performed an immunofluorescence assay against Ki67 protein, a proliferating cell marker (Figure 7, upper panel). As a result, in the case of the non-treated tumor, many proliferating cells exist around the outside of the tissue undergoing keratinization, while the number of proliferating cells were decreased in the tissues from the NCP and p-FAK/GNP groups, but they still exist. Interestingly, it was difficult to find proliferating cells in the tissues that treated NCP and p-FAK/GNP in combination. IHC for cleaved caspase 3 using each tissue was performed to see if each treatment method caused apoptotic cell death of SCC25 cells. As a result, no cleaved caspase 3 protein was observed in non-trained control cells, but an increase in apoptotic cleaved-caspase 3 was observed in tumor tissues treated alone with NCP and p-FAK/GNP, confirming that the two treatment methods induced apoptotic cell death. On the other hand, no cleaved caspase 3 protein was observed in tissues treated with NCP and p-FAK/GNP.

## 4. Discussion

Various studies are being actively conducted to use plasma for cancer treatment [18,19]. Numerous studies have shown that when CAP is used to treat cancer cells, apoptotic cell death is induced through an increase of intracellular ROS. In addition, many scientists reported that although the devices used are different, working elements such as OH radicals or NO, ONOO- and H2O2 that can dissolve in the medium during CAP treatment, act as key factors in the anti-cancer action of CAP [29]. Unfortunately, these CAP elements do not reach deep into the tissue when treated directly on the surface of the cancer tissue. To overcome this limitation of CAP, methods for treating cancer using plasma activated solutions (PAS) are also being studied [30,31]. However, since apoptotic cell death by increasing intracellular ROS is the main mechanism of action for both CAP and PAS-based cancer cell killing efficacy, it is difficult to show high anti-cancer efficacy and cancer selectivity compared to other anti-cancer drugs.

In our previous study, in order to amplify the selectivity and anti-cancer effect of CAP on melanoma, GNP conjugated with an antibody against p-FAK and NEU proteins was used in combination with CAP [25,32]. In this study, HaCaT (normal skin cells) and G361 human melanoma cell lines were pretreated with p-FAK/GNP and p-NEP/GNP and then treated with CAP using atmospheric air for 5 min. As a result, this combinational treatment immediately triggered cancer cell death, whereas the normal cells were not affected. However, since the CAP devices used in these studies are plasma generators using atmospheric air, high concentrations of ozone are inevitable. Therefore, there is a limitation in that the device cannot be used to treat OSCC located in the oral cavity, which is a part of the respiratory system.

Prior to this study, we developed NCP technology, a new plasma generation technology to safely use plasma in the oral cavity and confirmed that NCP does not generate ozone and effectively kills oral microorganisms at a very low temperature [27]. In addition, it was confirmed that when this technology is applied to dental cells, the toxicity and inflammatory reaction that can be induced by TEGDMA, one of the dental adhesives, can be suppressed [33], and bone formation can also be promoted [26]. In this study, we tested whether the combinational treatment of NCP with GNP targeting OSCC can kill OSCC cells immediately and elucidated its mechanism of action. First, to construct a GNP that selectively targets OSCC and oral cancer, the expression pattern of p-FAK protein in oral cancer cells was compared with that of normal cells. As a result (Figure 1), it was confirmed that p-FAK protein was highly expressed in SCC25 and YD-10B cells, which are OSCC cell lines, and MG63 cells, which are human osteosarcoma cells, compared to normal cells (HaCaT and HS68 cells). FAK is a non-receptor protein tyrosine kinase regulated by integrin signaling. However, since it is already well-known that FAK protein is overexpressed in many types of advanced cancer and plays an important role in the cancerous progression of tumors, the development of anti-cancer therapies targeting the FAK protein is also actively being carried out [34,35,36]. In particular, the Y397 residue of the FAK protein is the most essential phosphorylation site for protein activity [37], and its phosphorylation is involved in the carcinogenesis of cancers of the digestive organs [38]. In addition, Chiu et al. have reported that p-FAK is overexpressed in OSCC and plays an important role in the growth and invasiveness of OSCC [39]. Therefore, the results of this study suggest that when an antibody against p-FAK protein is conjugated to GNP, GNP not only has selectivity for OSCC, but can also reduce its growth and invasiveness.

Thereafter, the p-FAK/GNP was produced as per our previous report and tested to determine whether the combinational treatment of NCP with p-FAK/GNP induces the selective and immediate cell death of OSCC. The mere treatment of p-FAK/GNP had no significant effect on the viability of normal HaCaT cells and SCC25 cells at 2 h after the treatment, but the changes in cell shape were observed in SCC25 cells. This can be interpreted as a phenomenon mediated by the decreased function of FAK after the p-FAK/GNP treatment. Likewise, the mere treatment of cells with NCP alone also failed to induce the cell death of HaCaT and SCC25 cells. Other researchers have reported the anti-cancer effect of CAP on OSCC, but most of these studies monitored the anti-cancer activity of CAP at 24 h after CAP treatment. However, in Figure 2 of this study, the live/dead assay was performed immediately after the 5 min of NCP treatment and, therefore, the significant effect of NCP was not monitored. Interestingly, when p-FAK/GNP and NCP were co-treated, immediate cell death was observed only in SCC25 cells. Most anti-cancer activity using CAP causes apoptotic cell death, and it takes a long time to observe it. Therefore, our results suggest that the working mechanism for the immediate cell death of OSCC mediated by the combinational treatment of NCP with p-FAK/GNP might be different from that of CAP. In addition, the results of this study are consistent with the phenomenon that occurred when air plasma and p-FAK/GNP was co-treated on human melanoma cells in our previous study [25]. Since the chemical working elements of NCP, an argon plasma, and air plasma are different from each other [26,40], immediate cell death caused by combinational treatment of NCP and p-FAK/GNP may be caused by a physical element rather than a chemical element of NCP.

The change in the immediate killing efficacy of p-FAK/GNP+NCP according to the treatment method of NCP in this study also suggests the possibility that among many working elements of NCP, charged particles and electric fields might be key players for stimulating p-FAK/GNP on SCC25 cells. First, mere treatment of PAM (the media activated with NCP) on p-FAK/GNP-treated SCC25 cells showed a weaker anti-OSCC effect than p-FAK/GNP+NCP (Figure 3). This result suggests the fact that rather than chemical elements of NCP which can be melted in the media, physical elements of NCP, which directly stimulate p-FAK/GNP targeted in SCC25 cells, are key factors of the immediate OSCC killing effect of p-FAK/GNP+NCP. In addition, in Figure 4, the placement of electric grounded mesh during NCP treatment blocked about 78% of p-FAK/GNP+NCP-mediated immediate cell death, whereas the placement of dielectric mesh showed minor effects. This result also suggests that among the physical elements of NCP, electric fields and charged particles including argon ions and electrons (i.e., those that can be affected by grounded electric mesh) might be crucial for p-FAK/GNP+NCP-mediated cell death.

In the last part of this study, the anti-cancer effect of p-FAK/GNP+NCP against OSCC was tested in the mice cutaneous xenograft model using SCC25 cells. As a result of examining the external changes of the xenograft tumor (Figure 5), the p-FAK/GNP+NCP group showed the fastest decrease in tumor size compared to the mere treatment of NCP and p-FAK/GNP alone. In all three experimental groups, no wound on the skin surface was observed during the treatment period but, interestingly, only the p-FAK/GNP+NCP group showed bruise formation in the tumor tissue area. These results imply that the anti-OSCC activity of p-FAK/GNP+NCP is more immediate and stronger than the effect of NCP and p-FAK/GNP alone and, at the same time, these results also suggest that the specific mechanism of action might be different from others. Our results from the external and histological analysis using excised tumor tissue also showed that the anti-OSCC effect of p-FAK/GNP+NCP is fundamentally different from the effects of single treatment with NCP and p-FAK/GNP. When looking at the appearance and weight of the isolated tumor tissue at the end of animal experiments (Figure 6A), the size of tumors in the three experimental groups (NCP, p-FAK/GNP, p-FAK/GNP+NCP) were reduced to less than 50% of the control group, but there were no significant differences between the three experimental groups. Meanwhile, in our histological analysis data (Figure 6B), the tissue of the NCP and p-FAK/GNP group showed a decrease in cell density along with an increase in collagen when compared with non-treated controls. These results represent the fact that the mere treatment of NCP and p-FAK/GNP has an anti-OSCC effect, but the cancer cells still existed. Surprisingly, in the case of the p-FAK/GNP+NCP group, most tissue parts were filled with collagen, and few cells remained. In particular, the islands of malignant squamous epithelium, a characteristic tissue type of OSCC, were observed in the tumor tissues of the NCP and p-FAK/GNP groups, but this was not observed in the tissues of the p-FAK/GNP+NCP group. These results suggest that the combination treatment of NCP and p-FAK/GNP has the strongest anti-cancer activity. In the results of the experiment to examine the difference in cell proliferation and apoptotic cell death in tissues isolated from the three experimental groups, it was confirmed that the combined treatment of NCP and p-FAK/GNP had higher anti-cancer activity than the single treatment. In particular, active apoptotic cell death was observed in the tissue of the NCP and p-FAK/GNP groups, whereas it was not observed in the tissues of the p-FAK/GNP+NCP group. In line with the results of the in vitro experiments in this study, these in vivo data suggest that the combined treatment of NCP and p-FAK/GNP caused immediate cell death, not apoptotic cell death.

It is well-known that GNPs have a surface plasmon resonance (SPR) effect, which is the coherent oscillation of free electrons on the GNP’s surface, triggered by lights and energy [41]. Due to this property, many researchers reported that the combinational treatment of GNPs with various kinds of biomedical techniques such as lasers [42], radiations [43,44] and ultrasonics [45] can kill cancer cells by inducing necrosis and apoptosis of cancer cells depending on the size of the GNPs and the power of the GNP stimulating techniques they used [46]. In our study, p-FAK/GNP can be specifically bound to p-FAK proteins of the SCC25 cell surface; then, the NCP treatment might trigger an SPR effect of p-FAK/GNP, and this might induce the immediate death of OSCC cells in vitro and in vivo.

## 5. Conclusions

Numerous researchers have conducted various studies to use CAP for cancer treatment but, in most cases, several kinds of CAP devices induced apoptotic cell death by increasing the intracellular ROS of cancer cells, just as anti-cancer drugs and radiotherapy does. Therefore, the method for enhancing the selectivity and effectiveness of CAP’s anti-cancer activity is required. In addition, most CAPs have a problem with ozone, which can cause a big problem in the respiratory system, so there is a limit in that it cannot be used directly on oral cancer. To solve these problems associated with CAP, we adopted NCP, which can be safely used in the oral cavity in combination with p-FAK/GNP, which can target OSCC cells selectively. In this study, we confirmed that the combinational treatment of NCP and p-FAK/GNP can trigger the immediate cell death of OSCC cells selectively. Furthermore, we elucidated the fact that charged particles of NCP were main factors stimulating p-FAK/GNP, which bound to OSCC and induced immediate cell death, and the SPR effect might be involved in this phenomenon. Finally, we also confirmed that the mere treatment of NCP or the injection of p-FAK/GNP on xenograft mice models also had some anti-OSCC activity, but the combinational use of NCP and p-FAK/GNP was most effective. It is noteworthy that the mere treatment of OSCC tumor tissue with NCP or p-FAK/GNP alone induced apoptotic cell death, but the combinational treatment of NCP with p-FAK/GNP showed the strongest anti-OSCC effect in a non-apoptotic manner. Since there are many reports showing that p-FAK proteins were overexpressed in other types of malignant cancers besides OSCC and osteosarcoma, it is expected that the combinational treatment method of NCP and p-FAK/GNP can be applied to various types of cancers.

## Figures and Tables

**Figure 1 biomedicines-10-02259-f001:**
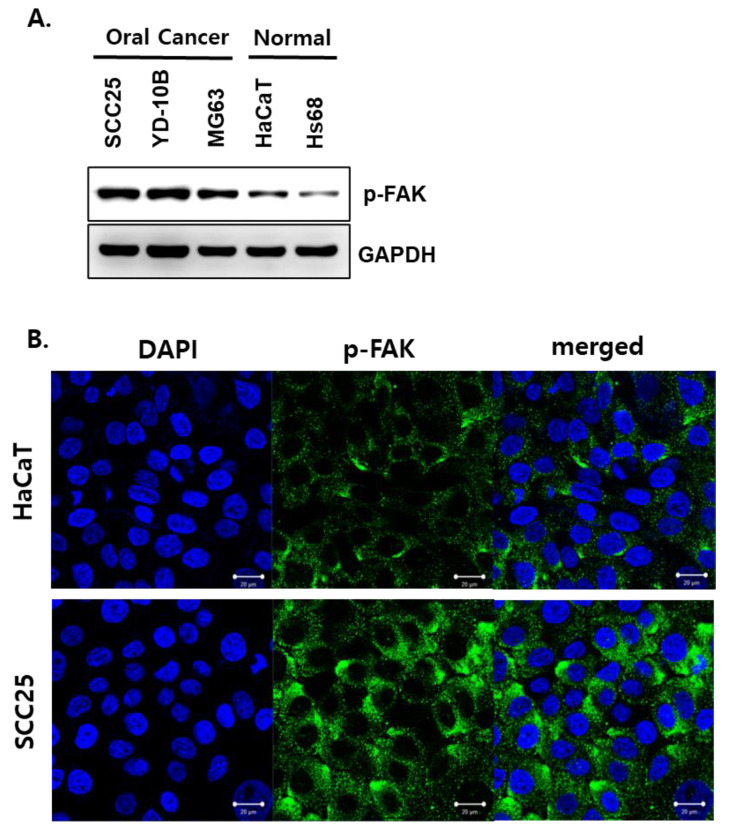
The p-FAK protein was overexpressed in oral cancers. (**A**) The basal expression level of p-FAK protein in oral cancer cells (SCC25, YD-10B and MG63) was compared with normal cells (HaCaT and Hs68) by performing Western blot assay. (**B**) The location and intensity of p-FAK protein in SCC25 and HaCaT cells was visualized through immunofluorescence assay. Scale bar: 20 μm.

**Figure 2 biomedicines-10-02259-f002:**
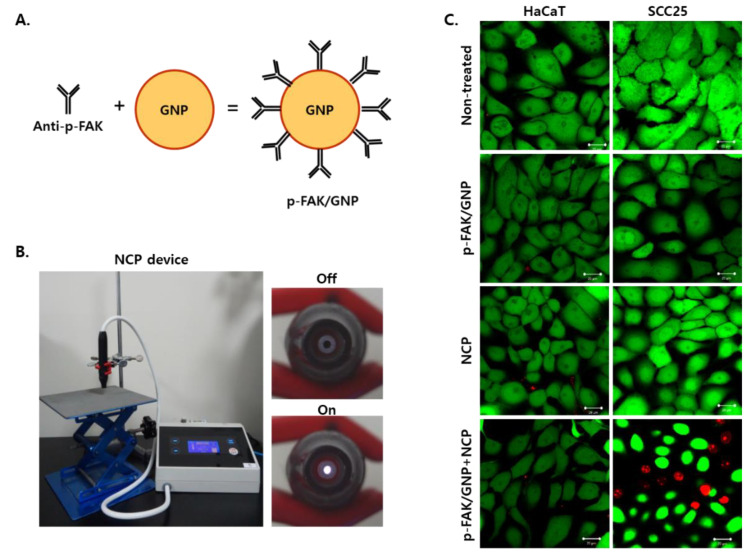
The combinational treatment of NCP with p-FAK/GNP selectively and immediately kills SCC25 cells. (**A**) A schematic diagram for the conjugation of p-FAK antibodies with GNP to form p-FAK/GNP. (**B**) Photographs showing the NCP device used in this study. (**C**) The results of the live/dead assay. The cells were treated with NCP, p-FAK/GNP and in combination, as described in the Materials and Methods section. The assay was performed immediately after the NCP treatment. The photographs are representative of the 3 independent experiments. Scale bar: 20 μm.

**Figure 3 biomedicines-10-02259-f003:**
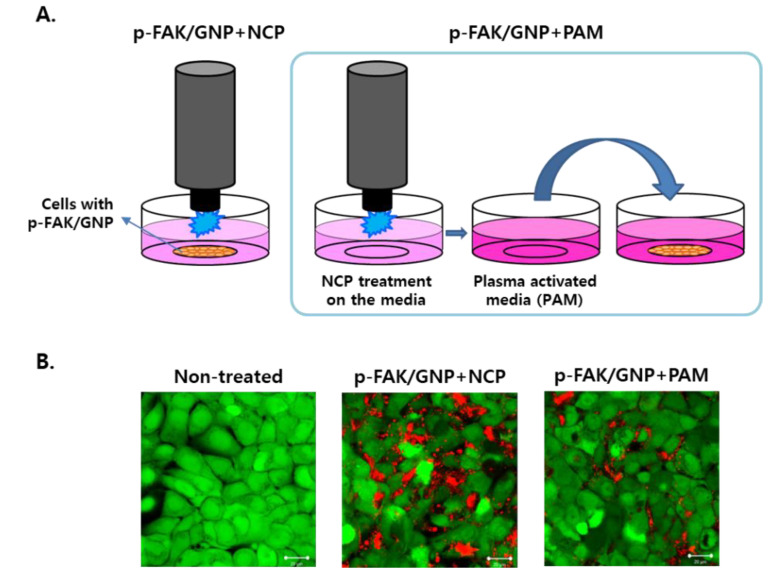
The direct treatment of NCP on p-FAK/GNP-treated cells induced immediate cell death, but plasma-activated media did not. (**A**) A schematic diagram describing the cell treatment methods for p-FAK/GNP+NCP and p-FAK/GNP+PAM. (**B**) The results of the live/dead assay depending on the NCP treatment methods. The SCC25 cells were pretreated with p-FAK/GNP 1 h before the NCP or PAM treatment. The photographs are representative of the 3 independent experiments. Scale bar: 20 μm.

**Figure 4 biomedicines-10-02259-f004:**
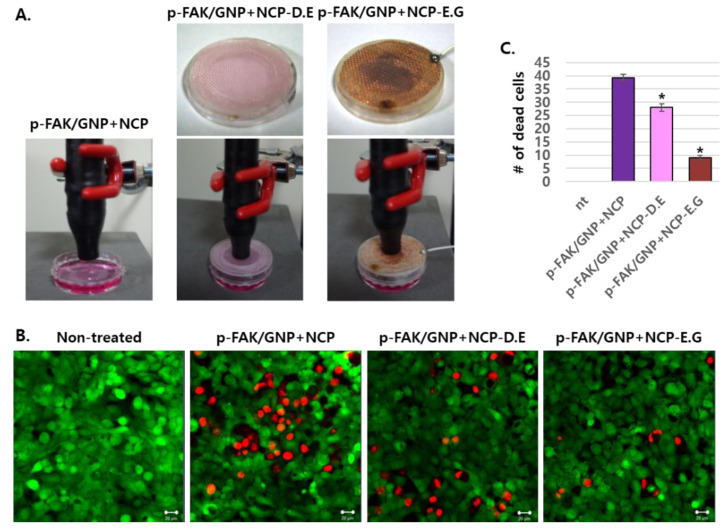
The p-FAK/GNP+NCP-mediated immediate cell death was blocked by electric grounded mesh. (**A**) Photographs showing the NCP treatment methods for p-FAK/GNP+NCP, p-FAK/GNP+NCP-DE (de-electric mesh) and p-FAK/GNP+NCP-EG (electric grounded mesh). (**B**) The results of the live/dead assay after the NCP treatment using 3 different methods. The SCC25 cells were pretreated with p-FAK/GNP for 1 h before the NCP or PAM treatment. The photographs are representative of the 3 independent experiments. Scale bar: 20 μm. (**C**) The number of dead cells in the photograph of (**B**) was counted, averaged and represented as a graph (*n* = 3), * *p* < 0.05.

**Figure 5 biomedicines-10-02259-f005:**
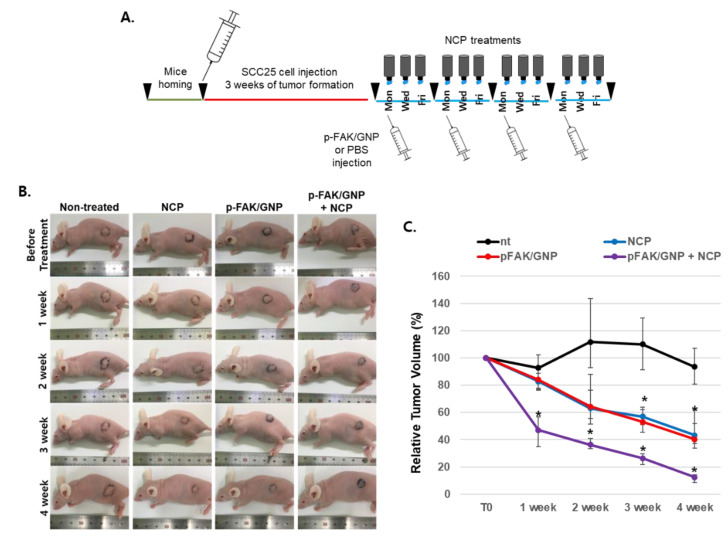
The anti-OSCC activity of the combinational treatment of p-FAK/GNP with NCP is faster and stronger than the single treatment of NCP or p-FAK/GNP in mice xenograft models. (**A**) A schematic diagram describing the animal experiments testing the anti-OSCC effect of p-FAK/GNP+NCP. After the tumor formation, p-FAK/GNP was injected once per week, and NCP was treated for 5 min on the tumor 3 times per week for 4 weeks. (**B**) The photographs showing external changes in the tumors in each group, which were taken before the treatment and after the last treatment of each week. The photographs shown are representative of each group (*n* = 4). (**C**) A graph showing the relative tumor volume change of each group (*n* = 4). The tumor volume of each group was measured after the photographs were taken. * *p* < 0.05.

**Figure 6 biomedicines-10-02259-f006:**
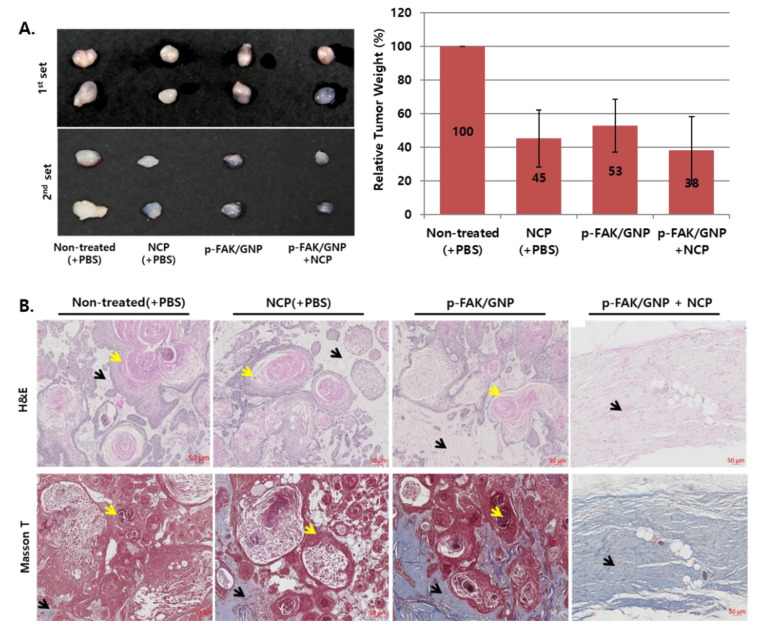
The tissues isolated from the p-FAK/GNP+NCP group were mainly composed of collagen, not SCC25 cells. (**A**) Photographs (left) showing the size and appearance of tissues isolated from each group, and a graph (right) showing the relative weight of isolated tissues. Data shown are the average of each group’s mice (*n* = 4). (**B**) The results of H&E and Masson’s T staining using the isolated tissues from each group. Data shown are representative of each group. Yellow arrow: keratinization; Black arrow: collagen. Scale bar: 50 μm.

**Figure 7 biomedicines-10-02259-f007:**
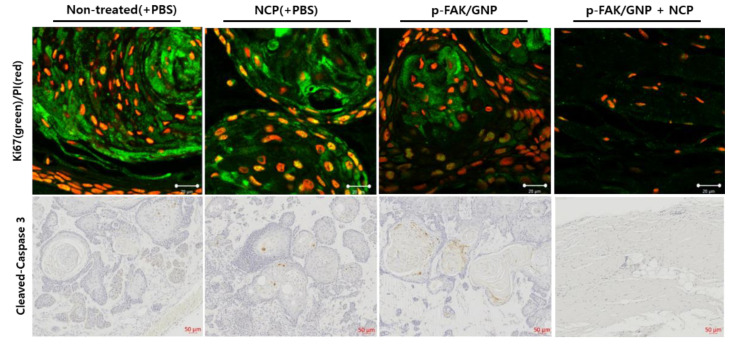
The proliferating and apoptotic cells are not detected in tissues isolated from the p-FAK/GNP+NCP group. (Upper panel) The results of the immunofluorescence assay against Ki67 protein using the tissues isolated from each group. The nuclear was counterstained with PI. Scale bar: 20 μm. (Lower panel) The results of IHC against cleaved caspase 3 protein. The nuclear was stained with hematoxylin. Scale bar: 50 μm.

## Data Availability

The data presented in this study are available upon request from the corresponding author.

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
