# Peer review of "Anti-Cancer Activity of the Combinational Treatment of Noozone Cold Plasma with p-FAK Antibody-Conjugated Gold Nanoparticles in OSCC Xenograft Mice"

_biomedicines, 2022, doi:10.3390/biomedicines10092259_

Round 1
Reviewer 1 Report (Previous Reviewer 1)
The revised manuscript has been improved and although it has some limitations it shows interesting results
Author Response
Dear the reviewer of this study,
It is our pleasure that our efforts was satisfactory to you.
Here, we checked our manuscript thoroughly, and confirmed that there were no spell error.
Sincerely yours,
Gyoo-Cheon Kim and Jeong-Hae Choi
Reviewer 2 Report (Previous Reviewer 2)
Dear Authors,
The manuscript biomedicines-1907532, entitled 'Anti-cancer activity of the combinational treatment of Nozone Cold Plasma with p-FAK antibody conjugated Gold Nanoparticles in OSCC Xenograft mice' is an improved version. The overall manuscript text is well written with good correlation of the data obtained by the authors. New references were added and the discussion paragraph was extended. Good quality figures are presented.
However, I would expect a better highlighted Conclusion paragraph ( extended).
After this small issue, I would recommend the manuscript for consideration to Biomedicine Journal.
MINOR REVISION
Author Response
Dear the reviewer of this study,
Thanks for your kind comment.
It was our pleasure that our responses to your comment was satisfactory to you.
As your comment, we modified the conclusion part, as below:
Numerous researchers have conducted various studies to use CAP for cancer treatment, but in most cases, several kinds of CAP devices induced apoptotic cell death by increasing the intracellular ROS of cancer cells, just as anti-cancer drugs and radiotherapy does. Therefore, the method for enhancing the selectivity and effectiveness of CAP’s anticancer activity is required. In addition, most CAPs have a problem with ozone, which can cause a big problem in the respiratory system, so there is a limit that it cannot be used directly on oral cancer. To solve these problems of CAP, we adopted NCP, which can be safely used in oral cavity, in combination with p-FAK/GNP which can targeting OSCC cells selectively. In this study, we confirmed that the combinational treatment of NCP and p-FAK/GNP can trigger immediate cell death of OSCC cells selectively. Furthermore, we elucidated the fact that charged particles of NCP was main factor for stimulating p-FAK/GNP which bound to OSCC and inducing immediate cell death, and SPR effect might be involved in this phenomenon. Finally, we also confirmed that the mere treatment of NCP or the injection of p-FAK/GNP on xenograft mice model also have some anti-OSCC activity, but the combinational use of NCP and p-FAK/GNP was most effective. It is noteworthy that the mere treatment of OSCC tumor tissue with NCP or p-FAK/GNP alone induced apoptotic cell death, but the combinational treatment of NCP with p-FAK/GNP showed most strong anti-OSCC effect in non-apoptotic manner. Since there are many reports showing that p-FAK proteins were overexpressed in other types of malignant cancer besides OSCC and osteosarcoma, it is expected that the combinational treatment method of NCP and p-FAK/GNP can be applied to various types of cancers.
We hope that our modified conclusion can be satisfactory to you.
Sincerely yours,
Gyoo-Cheon Kim and Jeong-Hae Choi
This manuscript is a resubmission of an earlier submission. The following is a list of the peer review reports and author responses from that submission.
Round 1
Reviewer 1 Report
This article disclaimed the combination of Nozone Cold Plasma (NCP) and gold nanoparticles conjugated with p-FAK antibody (p-FAK/GNP) in the therapy of Oral Squamous Cell Cancer (OSCC). The possible physical inhibitory effect on tumor cells by NCP was notable but required more proof and further mechanical investigation. However, the data in the article does not hold up to the conclusions; some biological assays are not well designed.
- In Figure 7, it is incredible that both NCP and p-FAK/GNP can cause apoptosis, respectively, but the combination of NCP and p-FAK/GNP induces negligible apoptosis.
- The identification of the vertical coordinates in Figure 3c should be the relative tumor volume rather than the tumor weight, and the percentage of tumor volume reduction is not consistent with the images of tumor tissues in Figure 3b.
- In Figures 2c, 3b, and 4b, dead cells should be quantified by additional methods like flow cytometry or CCK-8 assay, not just by cell staining with calcein AM and ethidium homodimer-1 mixture.
- In Figure 1, the confocal fluorescence imaging does not indicate the location of the p-FAK protein in the cell membrane. Co-staining of the p-FAK protein and cell membrane should be performed to confirm their co-localization.
- There are numerous confusing expressions and grammatical errors.
Reviewer 2 Report
Dear authors,
The manuscript biomedicines-1736541-peer-review-v1, entitled: ‘Anti-cancer activity of the combinational treatment of Nozone Cold Plasma with p-FAK antibody conjugated Gold Nanoparticles in OSCC Xenograft mice’ aims to verify whether Nozone Cold Plasma (NCP) source, which is designed for safe usage on oral cavities combined with gold NP, can instantly kill Squamous-Cell-Carcinoma (SCC-25) both in-vitro and in-vivo conditions, as well as to try to explain its mechanism on mice.
The authors concluded that, based on the presented results, their study reveals that the treatment with p-FAK/GNP or with NCP on xenograft tumor slowly decreased the size of tumor, but at the end of these experiments the tumor sized was about 50% in size. By combining these 2 treatments, only one week of p-FAK/GNP+NCP treatment was enough for reducing half of tumor size, moreover the authors found that even most of tumor tissue was vanished at the end. Their experimental results support the usage of this combined techniques of p-FAK/GNP+NCP as new methods for treating oral squamous cell carcinoma (OSCC).
The methods used for the investigation are well described, with figures that have clear schematics, as well argued. The ‘usual’ techniques and methods for such experiments are described, such as immunofluorescence, western blot, the formation of p-FAK/GNP, cell staining, the anti-OSCC activity evaluation of mice, histological assays as well as statistical analysis. However, the source of the plasma was extraordinarily briefly presented (page 3, paragraph 2.5). The overall manuscript text is well written, with good quality images and graphs.
The conclusions are short and objective.
However, I would make some modifications, for better visualization and readability:
- I expect that the authors make all the images & figures at least text-wide (bigger than in this manuscript version), for better visualization and understanding;
- Please insert more details on the plasma source, and proper reference (as reference 26 is not enough, maybe https://doi.org/10.1177%2F1535370221996655 or even https://doi.org/10.1038/s41598-022-11665-z ) on it’s characteristics, on the 2.5 paragraph, page 3, lines 126 tot135. I would expect info related to the plasma frequency, current intensity, or average / mean applied power. The Ar flowrate ( as working gas) was set to 1slm, as author stated; Is it optimum this flowrate, did you try different flowrates? Did other gases were tested also ( eg. He, N2, mixtures)?
- Did the plasma power was changed / varied in this study? Does it have a significant role on the described effects on OSCC? What about the excited plasma species? It would be of great help to the reader if an emission spectra of the plasma in contact with the surface, during treatment, would be included in the manuscript. Maybe the excited reactive species crated could play a role in these findings!?
- The conclusions to be expanded and more precisely defined;
By addressing the upper aspects, the manuscript can be considered for publication in the Biomedicines journal: MINOR REVISION.